# Unassisted and Carbon Dioxide-Assisted Hydro- and Steam-Distillation: Modelling Kinetics, Energy Consumption and Chemical and Biological Activities of Volatile Oils

**DOI:** 10.3390/ph15050567

**Published:** 2022-05-01

**Authors:** Sara El-Kharraf, Soukaïna El-Guendouz, Farah Abdellah, El Mestafa El Hadrami, Alexandra M. Machado, Cláudia S. Tavares, Ana Cristina Figueiredo, Maria Graça Miguel

**Affiliations:** 1Applied Organic Chemistry Laboratory, Faculty of Sciences and Technique, University Sidi Mohamed Ben Abdellah, Imouzzer, Fes 30000, Morocco; sara_kh_17@hotmail.fr (S.E.-K.); farah.abdellah1@gmail.com (F.A.); elmestafa.elhadrami@usmba.ac.ma (E.M.E.H.); 2Faculdade de Ciências e Tecnologia, C8, Campus de Gambelas, Universidade do Algarve, 8005-139 Faro, Portugal; soukaina.elguendouz@gmail.com; 3Centro de Estudos do Ambiente e do Mar (CESAM Lisboa), Faculdade de Ciências da Universidade de Lisboa, Biotecnologia Vegetal, DBV, C2, Piso 1, Campo Grande, 1749-016 Lisboa, Portugal; ialexam@gmail.com (A.M.M.); claudiasstavares@gmail.com (C.S.T.); acsf@fc.ul.pt (A.C.F.); 4Mediterranean Institute for Agriculture, Environment and Development, Faculdade de Ciências e Tecnologia, C8, Campus de Gambelas, Universidade do Algarve, 8005-139 Faro, Portugal

**Keywords:** antioxidant activity, enzyme inhibitory activity, carbon dioxide assisted distillation, distillation, extraction kinetics, chemical composition

## Abstract

The demand for more suitable eco-friendly extraction processes has grown over the last few decades and driven research to develop efficient extraction processes with low energy consumption and low costs, but always assuring the quality of the volatile oils (VOs). The present study estimated the kinetic extraction and energy consumption of simultaneous hydro- and steam-distillation (SHSD), and SHSD assisted by carbon dioxide (SHSDACD), using an adopted modelling approach. The two isolation methods influenced the VOs yield, chemical composition and biological activities, namely, antioxidant, anti-glucosidase, anti-acetylcholinesterase and anti-inflammatory properties. SHSDACD provided higher VOs yields than the SHSD at a shorter extraction time: 2.8% at 30 min vs. 2.0% at 120 min, respectively, for *Rosmarinus officinalis*, 1.5% at 28 min vs. 1.2% at 100 min, respectively, for *Lavandula angustifolia*, and 1.7% at 20 min vs. 1.6% at 60 min, respectively, for *Origanum compactum*. The first order and sigmoid model fitted to SHSD and SHSDACD, respectively, with R^2^ value at 96% and with mean square error (MSE) < 5%, where the k distillation rate constant of SHSDACD was fivefold higher and the energy consumption 10 times lower than the SHSD. The rosemary SHSD and SHSDACD VOs chemical composition were similar and dominated by 1,8-cineole (50% and 48%, respectively), and camphor (15% and 12%, respectively). However, the lavender and oregano SHSDACD VOs were richer in linalyl acetate and carvacrol, respectively, than the SHSD VOs. The SHSDACD VOs generally showed better capacity for scavenging the nitric oxide and superoxide anions free radicals as well as for inhibiting α-glucosidase, acetylcholinesterase, and lipoxygenase.

## 1. Introduction

Essential oils (EOs) are internationally defined as complex mixtures obtained by (1) hydro-, steam- or dry-distillation of a plant, or some of its parts, or by (2) expression, a mechanical process without heating, in the case of the epicarp of *Citrus* fruits [1]. EOs’ isolation techniques are thus included in a wider range of techniques that allow the obtaining of the volatile fraction of a plant, although the resulting extracts cannot be considered EOs.

CO_2_ supercritical fluid extraction, microwave-assisted extraction methods (compressed microwave-assisted hydrodistillation, microwave-accelerated steam-distillation, vacuum microwave hydrodistillation, solvent-free microwave extraction, microwave steam-distillation, microwave hydrodiffusion and gravity), headspace-based extraction techniques, e.g., static or dynamic headspace, solid phase micro-extraction (SPME), using a phase-coated fused-silica fiber as the adsorption medium, and ohmic-assisted steam-distillation [2,3,4] are some examples of volatiles’ extraction methodologies from aromatic plants. Nevertheless, some of these systems are only used in research laboratories since the volatiles are released to headspace or a phase-coated fused-silica fiber and are not isolated and stored. Supercritical fluid extraction uses a clean solvent (CO_2_) allowing high selectivity and purity of extracts, but the energy intensity during the compression and decompression stages can be a real problem, as well as the high cost of the system [5].

In the present work, in-lab built combined systems of simultaneous hydro- and steam-distillation (SHSD) and SHSD assisted by carbon dioxide (SHSDACD), Figure 1, were used for isolating the volatile oils (VOs) of *Rosmarinus officinalis* L. (rosemary), *Lavandula angustifolia* (lavender), and *Origanum compactum* Benth. (oregano). For the SHSDACD system, a chemical reactor that produced carbon dioxide (clean co-solvent) was used, Figure 1. Based on the reported increased physicochemical solvent proprieties of the carbon dioxide and water fluid mixture [6,7,8], it was hypothesized that this system would allow more efficient and homogeneous internal diffusion through the plant material, due to the high density and low viscosity of the carbon dioxide fluid generated and heated at 100 °C. In addition to the high fluid diffusion, a better solubility and faster release of the volatile compounds was achieved, accelerating the extraction process with shorter extraction time and lower energy consumption. Modelling of both systems was performed to evaluate and compare the extraction kinetic (extraction time, extraction diffusion coefficient, and recovery yields), in addition to their energy consumption. This approach at laboratorial scale allowed the collecting of important information on the experimental conditions and process behavior, offering a good preview of industrial scale. The chemical composition of the volatiles isolated by both systems was compared and some in vitro biological properties evaluated.

## 2. Results and Discussion

### 2.1. Effect of Carbon Dioxide Flow on the Distillation Process, Yield and Kinetics Modelling

The extraction times and the yields of *R. officinalis, L. angustifolia*, and *O. compactum* VOs, obtained with SHSD and SHSDACD, were dissimilar, Figure 2 and Table 1. Higher SHSDACD VOs yields were obtained with a distillation time < 30 min in comparison with those of SHSD with a distillation time > 100 min. The kinetic curves, Figure 2A–C, showed that the SHSDACD VOs recovery rate increased very fast at the beginning, reaching maximum yields at 30 min for *R. officinalis* (2.8%, *v*/*w*), at 28 min for *L. angustifolia* (1.5%, *v*/*w*), and at 20 min for *O. compactum* (1.7 %, *v*/*w*). This was significantly higher (*p*-value < 0.01) compared with the SHSD yields obtained for *R. officinalis* at 120 min (2.0%, *v*/*w*), *L. angustifolia* at 100 min (1.2%, *v*/*w*), and *O. compactum* at 60 min (1.6%, *v*/*w*), Table 1.

Oregano and rosemary SHSD VOs yields obtained in this study were similar to the EOs yields reported by some authors [9,10]. Conversely, Cutillas et al. [11] and Verma et al. [12] found a lower rosemary EO yield (0.8–1.5% after 3 h of extraction), depending on the plant growth stage. According to Perović et al. [13], the lavender EO yield could be >2% in water-to-flower ratio (hydromodule) optimal ratio. Innovative extraction processes, such as steam-distillation assisted by microwave or ultrasound, and supercritical CO_2_ also generated higher yields than the SHSDACD [10,14,15].

The extraction kinetic model of *R. officinalis*, *L. angustifolia*, and *O. compactum* VOs depended on the VOs amount extracted at any time, which can be defined as the volume of VO recovered per mass of the plant material, Figure 2. For both the SHSD and SHSDACD curve shapes there was a “washing” phase where the VOs were rapidly released from the plant: 78% of the rosemary leaf SHSDACD VO was released during the first 10 min, whereas with SHSD only 50% of the VO was released after 50 min. Likewise with lavender and oregano flowers, 80% to 90% of the VOs were released after 10 min. This may be related to the presence of external glandular structures in these species. Afterwards, the extraction occurred through a “diffusion” stage that involved two mechanisms: an “unhindered diffusion” involving the diffusion of the remaining VOs through the collapse (rupture) of the secretory glands; and an “hindered diffusion” from the intact reservoirs diffused through membranes and other barriers [16]. In *R. officinalis*, *L. angustifolia*, and *O. compactum* SHSD, the diffusion process was slower, only 15–30% of the VOs were recovered during 50 min before the equilibrium phase. However, in the SHSDACD diffusion process, a similar rate of VOs was quickly released in a short period < 20 min. The results suggest that the carbon dioxide flow mixed into the water steam affected the mass transfer by increasing the dissolving stage, and speeding glandular structures rupture, Figure 3.

The models previously mentioned, (1), (2), and (3), were applied to the experimental data, the goodness of fit of the kinetic models, and its parameters are depicted in Table 1. Depending on the mean square error (MSE) and the correlation coefficient R^2^, Table 1, the release mechanism for SHSD fitted to a first-order model with MSE value of 4% and R^2^ > 96%. Bousbia et al. [17] and Filly et al. [18] reported similar results for the release rate of rosemary and lavender volatiles, and that the simple first-order model is in a good agreement with it, considering only the diffusion phase into solid. However, no studies have reported on the kinetic simulation of *O. compactum* VO. In the present study, the sigmoid model showed a good fit to the SHSDACD experimental data, with a MSE value of 0% and R^2^ > 93–99%. These results agreed with those of the rosemary and lavender VOs extracted by microwave-assisted hydrodistillation, and microwave steam diffusion, respectively [19,20].

According to Tsimogiannis and Oreopoulou [21], the coefficient b, fast distillation coefficient, can be calculated through the y-intercept of the curves that equals ln(1-b), using the Equation (5) (Figure 4). That coefficient represents the VO portion that is removed during an initial period of distillation (theoretically at t = 0), which is characterized by a rapid increase in the VO yield. Therefore, a low b value indicates that the VO initially released from the surface glandular trichomes is slow. Moreover, a negative b value was found for all samples obtained by SHSDACD, Table 1, which suggests an initial inhibition of distillation. According to Tsimogiannis and Oreopoulou [21], negative b values can be caused by relatively high quantities of non-volatiles on the surface glandular trichomes, although they also refer to the need for a better study of the behavior explaining these deviations. To what extent the presence of a dense non-glandular indumentum slows down the initial stages of the distillation process also deserves attention. The k distillation rate constant value was 1000 times higher for the oregano VO than the remaining samples, that resulted from the fast diffusion of VO trapped in glandular trichomes.

### 2.2. Evolution of Energy Consumption during the SHSD and SHSDACD Extraction

To evaluate the difference in the consumption energy between the SHSD and SHSDACD process, the Equation (6) was adapted to the simulation extraction kinetic model previously studied for each process, Figure 5.

SHSDACD was an efficient extraction process with less extraction time in comparison with the SHSD. SHSDACD allowed the recovery of a high quantity of VOs in < 30 min and simultaneously reduced the extraction time and energy consumption, Table 1. With SHSDACD, the energy consumption was 10 times lower than SHSD for the rosemary and lavender VOs, suggesting that the solvent used provided better and fast heating of the plants, as well as better affinity and VOs’ dissolution than SHSD. In the case of the oregano VO, SHSDACD needed more energy than for the rosemary and lavender which might be explained by the initial inhibition of distillation previously mentioned, where it had the lowest fast distillation coefficient (b = −0.26), yet was still better than SHSD process. Similar results have been found for different conventional distillation methods intensified by solvent supercritical point, ohmic, microwave, or ultrasonic, etc. revealing that the input of energy increases the volatiles’ diffusion within a short extraction time, and leads to low energy consumption [22,23]. Meziane et al. [24] reported that microwave-assisted hydrodistillation consumed 2–10 less energy than hydrodistillation alone. Boukroufa et al. [25] and Desai and Parikh [26] reported similar results. 

### 2.3. Scanning Electron Microscopy Observations

SEM of *L. angustifolia* flowers, prior to distillation, showed intact glandular trichomes (Figure 3A). SEM images after SHSDACD and SHSD showed no major differences, between both distillation processes (Figure 3A,B) and all glandular cells evidenced collapsing. Likewise, similar SEM features were observed with *R. officinalis* and *O. compactum*.

### 2.4. Chemical Composition of VOs

In the lavender SHSD and SHSDACD VOs, 37 and 44 compounds were identified, constituting about 98.1% and 97.0%, respectively, Table 2. The main SHSD and SHSDACD VOs components were linalool (20.5% and 14.2%), linalyl acetate (13.1% and 25.5%), camphor (16.5% and 15.1%), 1,8-cineole (16.1% and 13.1%), and borneol (10.4% in both cases). These results agree with those reported for HD *L. angustifolia* essential oil from Morocco [27] and Italy [28], and for HD *L. latifolia* from Spain [29].

Twenty-eight to thirty-one components were identified in the oregano VOs, obtained by SHSD and SHSDACD, respectively, representing 98.1% and 96.4% of the total amount of VOs. SHSD and SHSDACD VOs were dominated by thymol (37.8% and 10.5%), *p*-cymene (27.3% and 25.0%), γ-terpinene (9.8% and 0.6%), α-terpineol (3.3% and 19.3%), and carvacrol (5.5% and 17.9%), Table 2. The chemical composition of the oregano VOs obtained by both extraction methods are according to those reported by Aboukhalid et al. [30] and Bouyahya et al. [31]. Both works showed chemical variability depending not only on the harvesting period, but also on this species’ high chemical polymorphism.

A total of 24 and 22 compounds were identified in the rosemary VOs obtained by SHSD and SHSDACD, respectively, representing more than 99.0% of identification, Table 2. The main components, that is, the components that were ≥5% in at least one of the two VOs, were 1,8-cineole (50.2% and 48.2%), camphor (15.3% and 11.7%), α-pinene (10.8% and 9.9%), borneol (3.7% and 5.2%), and β-caryophyllene (0.1% and 5.6%). Due to this last difference, the relative amount of sesquiterpenes was higher in the SHSDACD VOs, Table 2. The present results are in accordance with HD-obtained essential oils of rosemary from Morocco [34,35] while Chinese [19,36], and French rosemary essential oils presented additionally verbenone (2.0–6.6%, respectively) [37]. Conversely, a Turkish HD essential oil showed *p*-cymene (44.0%), linalool (20.5%), and γ-terpinene (16.6%) as the main components [38].

### 2.5. Antioxidant Activity

The comparative study of the antioxidant abilities of SHSD and SHSDACD *R. officinalis, L. angustifolia,* and *O. compactum* VOs was carried out using diverse methods: DPPH, nitric oxide, and superoxide free radical-scavenging activities, Table 3.

With DPPH assay, the lowest IC_50_ values were obtained with SHSDACD VOs, Table 3, whereas the strongest free radicals scavenging ability was observed for *O. compactum* VOs (IC_50_ at 0.01 mg/mL), much better than the BHT standard [39] (IC_50_ = 0.23 mg/mL). This VO activity was significantly different from the SHSD VO (*p*-value < 0.05) which may be attributed to the different *p*-cymene, thymol, and carvacrol ratio in SHSDACD VOs. Mastelić et al. [40] reported that thymol had stronger antioxidant activity than carvacrol, but Bouyahya et al. [31] and Bouhdid et al. [41] estimated that carvacrol-rich oregano VO was a potent antioxidant agent (IC_50_ < 0.01 mg/mL). Nevertheless, other studies showed no significant difference between these two isomers’ activities [42].

SHSD and SHSDACD *R. officinalis* VOs expressed a moderate scavenging capacity (3.1 mg/mL and 1.1 mg/mL) compared to *L. angustifolia* Vos (4.9 mg/mL and 3.5 mg/mL), probably due to the high 1,8-cineole and camphor content, despite 1,8-cineole having been reported as having a markedly lower IC_50_ value (5.7 ± 1.5 μg/mL) [43]. On the other hand, Selmi et al. [44] showed that the caryophyllene-rich rosemary VO had a stronger antioxidant ability with IC_50_ = 221.4 ± 4.3 μg/mL, than the rosemary VO with 1.8-cineole and α-pinene as its major components.

Lavender SHSDACD VOs, containing high levels of linalyl acetate and oxygen-containing sesquiterpenes, showed a low IC_50_ value, Table 2. Likewise, Kokina et al. [45], reported that linalyl acetate-rich lavender essential oil exhibited better antioxidant activity.

SHSDACD VOs showed significantly higher capacity for scavenging nitric oxide (*p* < 0.05), particularly the *O. compactum* VOs (IC_50_ = 0.05 mg/mL), which were fourfold lower than the remaining samples, although higher than the reference curcumin (IC_50_ = 0.01 mg/mL) [39]. Karimian et al. [46] reported that carvacrol and *p*-cymene-rich essential oils were better NO radical scavengers than those with thymol, *p*-cymene, and α-terpinene. The scavenging capacity of oregano VOs may result from the synergic effect between carvacrol and *p*-cymene. *R. officinalis* and *L. angustifolia* SHSD VOs did not permit the determination of the IC_50_ values, only the inhibition percentages (IP) (IP = 21.14 ± 1.18% and 29.36 ± 0.26%), depending on their main components such as 1,8-cineole, linalyl acetate, and linalool, that were previously shown to have no activity [47].

The best capacity for scavenging superoxide anion free radicals was found for *L. angustifolia* (IC_50_ = 0.21 mg/mL) and *O. compactum* (IC_50_ = 0.55 mg/mL) SHSDACD VOs, although the capacity was moderate when compared to the standard ascorbic acid (IC_50_ = 0.01 mg/mL) [39]. These VOs had relative high levels of linalyl acetate and carvacrol, respectively, then those obtained by SHSD, which may explain their action against the superoxide anions radicals. Lavender SHSD VOs, with linalool, 1,8-cineole, and camphor as the major compounds, showed IC_50_ values similar to those reported by Aazza et al. [42]. Linalool, ad 1,8-cineole, and camphor have shown poor inhibitory capacities [48,49].

SHSD *R. officinalis* VOs, rich in 1,8-cineole, and camphor, did not present good activity, Table 3, in comparison with SHSDACD VOs. Therefore, the differences observed in the NO and superoxide anion scavenging activities may depend on the difference between SHSD and SHSDACD VOs chemical composition. Moreover, the slight variation in the chemical composition can generate a considerable change in the antioxidant activity of the samples, confirming the importance of the synergistic or antagonistic interaction among the bioactive compounds of VOs.

### 2.6. Anti-Enzymatic Activity

α-Glucosidase is a key enzyme in the final step of oligosaccharide hydrolyzation, which is important in the regulation of glucose level in blood. Some currently used α-glucosidase inhibitors, such as acarbose, were found to promote negative side effects, such as abdominal distension, flatulence, and diarrhea [50], reinforcing the need for alternative natural agents. Oregano and rosemary SHSDACD VOs showed the lowest IC_50_ (3.24 and 8.57 µg/mL, respectively) and significantly different from the SHSD VOs (*p* < 0.05), Table 4. Those values were lower, therefore better, than that of acarbose (IC_50_ 0.014 mg/mL) [39]. The best inhibitory activity of oregano and rosemary SHSDACD VOs may be attributed to the relative high levels of carvacrol and α-terpineol, Table 2. Lavender SHSD VOs showed the poorest inhibitory capacity (IC_50_ = 721.07 µg/mL). This VO had lower linalyl acetate and higher linalool content than lavender SHSDACD VOs. Considering that these two compounds did not show anti-α-glucosidase activity in previous studies [39], the inhibitory activity cannot be attributed solely to these compounds but, instead, to the whole VO. The anti-α-glucosidase activity of lavender VOs obtained by both extraction methods was within the range found by Najibullah et al. [51]. SHSD rosemary VOs, showed higher IC_50_ (10 μg/mL) than the value reported by Ahamad et al. [52] for the EO recovered by hydrodistillation with verbenone as main component, yet the reported IC_50_ was closer to that found for SHSDACD VOs in the present work.

A natural alternative to treatment of Alzheimer’s disease, in which a decline in cognitive abilities occurs, may include the use of EOs to prevent the reduction of acetylcholine in the brain, inhibiting the acetylcholinesterase activity. The strongest enzymatic inhibitory activity was observed for both oregano VOs, immediately followed by lavender SHSD VO (IC_50_ = 10.66, 71.47, and 83.25 µg/mL, respectively), Table 4, yet still higher than the galantamine IC_50,_ reported by El-Kharraf. et al. [47]. Some authors [47,53,54] demonstrated that α-pinene and limonene were responsible for inhibition capacity, either alone or in combination with the remaining VOs constituents. López et al. [55] reported that SD oregano essential oil, as well as their single main compounds, carvacrol and thymol, presented a suitable potential against acetylcholinesterase (IC_50_ = 0.124, 0.113, and 0.625 mg/mL, respectively), and all these values were higher (lower activity) than those found in the present work. Despite the result found for the lavender SHSD VO in the present work, Table 4, Ferreira et al. [56] reported that 1 mg/mL of Portuguese lavender EO can only inhibit 39.5 ± 8.6% of acetylcholinesterase activity. SHSD rosemary VO expressed the poorest anti-AChE ability. The IC_50_ values for both of the rosemary VOs (513.85 and 349.18 µg/mL) were superior to those found by Cutillas et al. [11] for Spanish rosemary. The authors associated the anti-acetylcholinesterase activity to rosemary’s VO 1,8-cineole and 3-carene content.

All of the SHSDACD VOs presented better activity than the remaining SHSD samples. The IC_50_ values for those VOs were even lower (then better) than the control (NGDA) (0.04 mg/mL) [39]. According to Carrasco et al. [29], linalool, camphor, *p*-cymene and limonene had good anti-lipoxygenase activity. Frum and Viljoen [57] also reported good activity for limonene and β-caryophyllene, but not for 1,8-cineole, only when combined with limonene. This reveals a potential synergic effect between them, showing the importance of interaction among the compounds that constitute an essential oil. El-Kharraf et al. [39] also reported good anti-lipoxygenase activity of *p*-cymene. The inhibitory activity of linalool reported by Carrasco et al. [29] was not found by other authors [58] along with other alcohols such as geraniol and 2-phenylethyl alcohol. According to Frum and Viljoen [57] this inability could be attributed to their electronegative hydroxyl function that would prevent them reaching the enzyme active site. Nevertheless, other alcohols tested by Baylac and Racine [58] showed relatively high inhibitory activity of lipoxygenase, being all of them sesquiterpenoids (α-bisabolol, *trans*-nerolidol, and farnesol). These results can be considered contradictory, but Frum and Viljoen [55] also pointed out that as a result of 5-lipoxygenase enzymatic reaction being stereospecific, the response of the two isomers for the same terpenoid can be different.

## 3. Materials and Methods

### 3.1. Plant Material

*Rosmarinus officinalis* leaf, and the flowers of *Lavandula angustifolia* and *Origanum compactum*, were harvested, in May and June 2017, from their wild habitat in Morocco, in Talsint, Figuig province (32°51′54″ N, 3°27′02″ W), Oulmes, Khémisset province (33°25′48″ N, 6°0′36″ W), and Jbel EL Aalam, Tetouan-Tanger province (35°18′36″ N, 5°30′36″ W). Talsint and Oulmes regions both have a warm Mediterranean climate with a dry summer, with average temperatures and precipitation of 20.1 °C, 157 mm and 20.1 °C, 207 mm, respectively. Jbel EL Aalam region is characterized by a humid Mediterranean climate, with average temperature and precipitation of 22 °C and 2000 mm, according to the Köppen–Geiger classification.

The three species were identified by Prof. Abdellah Farah and codes were assigned as *R. officinalis*: Ref. LCOA-FST 018/2017, *L. angustifolia:* Ref. LCOA-FST 020/2017, and *O. compactum*: Ref. LCOA-FST 012/2017.

### 3.2. SHSD and SHSDACD Extraction Methods

The volatile oils (VOs) were extracted using two different methods, Figure 1, a simultaneous hydro- and steam-distillation (SHSD) and SHSD assisted by carbon dioxide (SHSDACD). The SHSD was performed in a distillation apparatus with slight modifications, namely by adding a biomass flask, Figure 1A. The SHSDACD consisted of a reactor that produced a flow of CO_2_ by a chemical reaction, connected to the 1000 mL boiling water containing twin-neck round-bottom flask, linked to the biomass flask, on the top of which a distiller and a condenser were connected, Figure 1B,C. The carbon dioxide in the reactor was produced by the chemical reaction of concentrated hydrochloride acid (6 M) and calcium carbonate CaCO_3_ as the following equation: CaCO_3_ + 2HCl → H_2_O + CO_2_ + CaCl_2_, where 1 g of the reagent released 224 mL of CO_2_. One hundred grams of dried plants were placed in the biomass flask and heated by convection with a mix of steam and carbon dioxide. The heated fluid then diffused through the plant material carrying out the volatile oils. The low density and immiscibility of the VOs allowed their separation from the hydrolate in the distiller unit. The distillation conditions were the same for the SHSD and SHSDACD: 100 g of plant material with 17% humidity, 1000 mL of water, 100 °C temperature, 0.97 L/h distillate flow, 224 mL/g CO_2_ flow, leaves of rosemary, and flowers of lavender and oregano. The obtained volatile oils were dried using anhydrous sodium sulphate and stored at 4 °C until use in the upcoming experiments.

### 3.3. Kinetics of the Volatile Oil Extraction

To study the kinetic behavior of the SHSD and SHSDACD extraction processes, three different models, under equilibrium conditions, were selected to fit release kinetic curves of VOs [24,59,60,61].

#### 3.3.1. First-Order Kinetic Model

The first order kinetic model described by Samadi et al. [59] has been established on diffusion phenomenon through solid using the Fick’s second law. It is represented by Equation (1):(1)$$\frac{{\mathrm{dY}}_{t}}{\mathrm{dt}}=K\left(Y_{e}-Y_{t}\right)$$

The Y_t_ (mL of VO/100 g of plants %) and Y_e_ (mL of VO/100 g of plants %) are the yield of VOs at any time (min) or at the equilibrium, respectively, t: time in min. K: rate constant.

The Equation (1) integrated at the initial condition Y_t_ = 0 at t = 0 and the boundary condition Y_t_ = Y at t = t to a nonlinearized Equation (2):(2)$$Y_{t}=Y_{e}\left(1-e^{-\mathrm{kt}}\right)$$

#### 3.3.2. Adsorption Kinetic Model

The empirical model proposed by Babu and Singh [60] fitted the desorption phenomenon in the case of exogenous VO deposits or broken cell walls. The model is generally described as shown in Equation (3):(3)$$\frac{Y_{t}}{Y_{\infty }}=\frac{t}{b+t}$$
where Y_t_ (mL/100 g of plants %) and Y_∞_ (mL/100 g of plants %) were the VO recovered amount at any time and at infinite time, respectively; b=K1K2 is the constant depending on two parameters as the K_1_ is the extraction rate constant, and K_2_ is the extraction capacity constant.

#### 3.3.3. Sigmoid Model

The kinetic extraction process of VOs by SHSD and SHSDACD have been described by a non-linear equation deviated from the above-mentioned models according to Meziane et al. [24] and Stanisavljević et al. [61], which reported that the asymptotic value collected can be described by a Boltzmann sigmoid curve as shown in Equation (4):(4)$$\frac{Y_{\infty }-Y_{t}}{Y_{\infty }}=\left(1-b\right)e^{-\mathrm{kt}}$$
where Y_t_ (mL of VO/100 g of plants %) and Y_∞_ (mL of VO/100 g of plants %) were the VO recovered amount at any time and at infinite time, respectively; b is the constant depending on various operating parameters as steam flow rate and plant material mass.

To calculate the k and b of equation, the linearized form can be used, as shown in Equation (5):(5)$$\ln\frac{Y_{\infty }-Y_{t}}{Y_{\infty }}=\ln\left(1-b\right)-\mathrm{kt}$$

### 3.4. Evolution of Energy Consumption during the SHSD and SHSDACD Methods

The energy consumption was estimated using Equation (6), where P is the applied power and t is the extraction time [24].
(6)$$E=\frac{P\times t}{60,000}$$

P(W) is the source power applied, t (min) is the time of extraction, and E (kWh) is the energy.

The energy consumption during the extraction process could be estimated by expressing the time extraction at each instant from kinetic previous models (1), (2), and (3) where the equations are represented as follows in Equations (7)–(9), respectively:(7)$$\mathrm{First}\;\mathrm{order}\;\mathrm{model}:\;E_{\mathrm{FO}}=-\frac{P}{60,000K}\ln[1-\frac{Y_{t}}{Y_{e}}]$$

The Y_t_ (mL of VO/100 g of plants %) and Y_e_ (mL of VO/100 g of plants %) are the yield of VO at any time (min) or at the infinity time, respectively, t (min) is the time, K is the rate constant.
(8)$$\mathrm{Adsorption}\;\mathrm{model}:\;E_{A}=\frac{P\times b}{60,000K}\times [\frac{\frac{Y_{t}}{Y_{\infty }}}{1-\frac{Y_{t}}{Y_{\infty }}}]$$
where Y_t_ (mL of VO/100 g of plants %) and Y_∞_ (mL of VO/100 g of plants %) were the VO recovered amount at any time and at infinite time, respectively; b is the constant depending on two parameters as the K_1_ is the extraction rate constant, and K_2_ is the extraction capacity constant.
(9)$$\mathrm{Sigmoid}\;\mathrm{model}:\;E_{A}=-\frac{P}{60,000K}\ln[(1-\frac{Y_{t}}{Y_{\infty }})\times \frac{1}{\left(1-b\right)}]$$
where Y_t_ (mL of VO/100 g of plants %) and Y_∞_ (mL of VO/100 g of plants %) were the VO recovered amount at any time and at infinite time, respectively; b is the constant depending on various operating parameters as steam flow rate and plant material mass.

### 3.5. Model Validation

The goodness of fit of the kinetic models used with the experimental data of SHSD and SHSDACD was evaluated using mean square error (MSE) between the simulation of the models and experimental data in which the calculated sum of squares errors divided by the length of the actual data period is shown in Equation (10).
(10)$$\mathrm{MSE}=\left[\frac{\sum_{i=1}^{\mathrm{NT}}\left(y_{\mathrm{th}}-y_{\exp}\right)}{n}\right]$$
where y_th_, y_exp_, and n are the model data, experimental data and the length of the actual data period, respectively.

### 3.6. Scanning Electron Microscopy (SEM)

The morphology and the secretory gland sizes were determined by scanning electron microscopy. The plant material, prior to and after distillation, was dehydrated by immersing in dry ethanol, then the solvent was changed twice at 30 min intervals and then placed in a desiccator. After that they were placed on a Peltier board for observation at 25 °C and 90 Pa, using a Quanta 200 FEI equipped with EDAX probe (innovation and technology transfer center CURI, Fez, Morocco), 0.5 at 30 KV, Det LFD, 3.5 nm, from 250× to 400×, between 15.00 and 20.00 KV.

### 3.7. Chemical Analysis of Volatile Oils

Gas chromatography (GC) and Gas chromatography-mass spectrometry (GC-MS) analyses were performed according to that previously reported by Machado et al. [62] and ISO 7609 [63]. 

Gas chromatographic analyses were performed using a Perkin Elmer Clarus 400 gas chromatograph (Perkin Elmer, Shelton, CT, USA) equipped with two flame ionization detectors (FIDs), a data handling system, and a vaporizing injector port into which two columns of different polarities were installed: a DB-1 fused-silica column (polydimethylsiloxane, 30 m × 0.25 mm i.d., film thickness 0.25 µm; J & W Scientific Inc., Rancho Cordova, CA, USA) and a DB-17HT fused-silica column ((50% phenyl)-methylpolysiloxane, 30 m × 0.25 mm i.d., film thickness 0.15 µm; J & W Scientific Inc.). The oven temperature was programmed, 175 °C, at 3 °C/min, subsequently at 15 °C/min up to 300 °C, and then held isothermal for 10 min; injector and detector temperatures, 280 °C and 300 °C, respectively; carrier gas, hydrogen, adjusted to a linear velocity of 30 cm/s. The samples were injected using split sampling technique, ratio 1:50. The volume of injection was 0.1 µL of a *n*-pentane-volatile oil solution (1:1). The percentage composition of the volatiles was computed, by the normalization method from the GC peak areas, calculated as mean values of two injections, from each sample, without using the response factors, in accordance with ISO 7609 [63].

#### Gas Chromatography-Mass Spectrometry: GC-MS Analysis

The GC MS unit consisted of a Perkin Elmer Clarus 600 gas chromatograph, equipped with DB 1 fused-silica column (30 m × 0.25 mm i.d., film thickness 0.25 µm; J & W Scientific, Inc.), and interfaced with a Perkin-Elmer 600T mass spectrometer (software version 5.4.2.1617, Perkin Elmer, Shelton, CT, USA). Injector and oven temperatures were as above; transfer line temperature, 280 °C; ion source temperature, 220 °C; carrier gas, helium, adjusted to a linear velocity of 30 cm/s; split ratio, 1:40; ionization energy, 70 eV; scan range, 40–300 u; scan time, 1 s. The identity of the components was assigned by comparison of their retention indices, calculated in accordance with ISO 7609 [64], relative to C_9_ C_17_
*n*-alkane indices and GC MS spectra from a lab-made library, created with reference essential oils, laboratory-synthesized components, laboratory-isolated compounds and commercially available standards.

### 3.8. Free Radical Scavenging Assays

#### 3.8.1. 2,2-Diphenyl-1-Picrylhydrazyl (DPPH) Free Radical-Scavenging Assay

DPPH radical free scavenging capacity was assessed according to the method described by Navanesan et al. [64], with slight modifications. Briefly, 250 µL of VOs concentrations ranging between 10^−3^ to 10 mg/mL were added to 750 µL of DPPH solution (1 mM) and directly incubated for 30 min at 25 °C. The absorbance of the DPPH radical decreasing was measured at 517 nm using an Ultraviolet-visible spectrophotometer. The radical inhibition percentage was calculated with the equation: Inhibition (%) = [(A_blank_ − A_sample_)/A_blank_ × 100], where A_blank_ and A_sample_ are the absorbance of the control and the absorbance of the sample, respectively. The sample and positive control concentration providing 50% inhibition (IC_50_) was achieved by plotting the inhibition percentage against samples concentrations.

#### 3.8.2. Nitric Oxide Free Radical-Scavenging Assay

The assay was conducted as described by El-Guendouz et al. [65] with minor modifications. A volume of 150 µL of various concentrations of samples were added to 150 µL of sodium nitroprusside solution (10 mM in PBS) then incubated for 60 min at 25 °C. Then, 100 µL of Griess reagent was added to the previous preparation and directly was measured in an UV spectrophotometric Microplate reader at 548 nm. The inhibition percentage was calculated using the formula: Inhibition (%) = [(A_blank_ − (A_sample_ − A_correction_)/(A_blank_)] × 100, where A_blank_ is the absorbance of sodium nitroprusside with ethanol 96%, A_sample_ and A_correction_ were the absorbance of the sample and positive control with and without sodium.

#### 3.8.3. Superoxide Anion Free Radical Scavenging Assay

Superoxide anion scavenging of VOs was estimated using a nonenzymatic phenazine methosulfate-nicotinamide adenine dinucleotide (PMS-NADH) system that generated those radicals by oxidation of NADH, according to El-Guendouz et al. [65]. Briefly, 25 µL of different concentrations of samples were combined with 25 µL of nitrotetrazolium blue chloride (43 µM NBT) and 50 µL of NADH (2.7 µM) in a phosphate buffer (19 mM, pH = 7.4). Then, 25 µL of PMS (166 µM) was added to the previous solution to generate the superoxide anion after 10 min incubation period at 25 °C. The absorbance reading was measured at 560 nm in a microplate reader, the anion scavenging was estimated using the following equation: Inhibition (%) = [(A_blank −_ A_sample_)/A_blank_ × 100], where A_blank_ and A_sample_ were the absorbance of the control and the absorbance of the sample, respectively. The (IC_50_) value was determined as reported previously.

### 3.9. Enzymatic Assays

#### 3.9.1. α-Glucosidase Inhibition Assay

The α-Glucosidase inhibitor capacity of VOs was carried out according to El-Guendouz et al. [66]. Briefly, 25 µL of various concentration of samples was added to 30 µL of yeast α-glucosidase (2.4 U/mL in phosphate buffer pH = 6.8) and incubated for 10 min, then 100 μL of *p*-nitrophenyl-β-d-glucopyranoside solution (0.5 mM) was mixed with the previous preparation. The solution was incubated a second time at 25 °C for 30 min, 50 μL sodium carbonate solution (0.4 mM) was added to stop the reaction. The absorbance measurement was performed at 405 nm. The percentage of α-glucosidase inhibition was calculated using the following equation: Inhibition = [(A_blank_ − (A_sample_ − A_correction_)/(A_blank_)] × 100; where A_blank_ is the absorbance of the control. A_sample_ and A_correction_ were the absorbance of the sample with and without the enzyme, respectively.

#### 3.9.2. Acetylcholinesterase Inhibition Assay

The acetylcholinesterase inhibitory ability of VOs was evaluated in a microplate by the spectrophotometric method [66]. A volume of 25 µL of different concentrations of VOs was added to 60 µL enzyme (0.28 U/mL in Tris-HCL buffer 0.1 M, pH = 8), then incubated for 15 min. A volume of 50 µL of substrate acetylcholine was added. After that, 125 µL of 5,5′-dithiobis-2-nitrobenzoic acid (59 mg DTNB in 50 mL buffer) were mixed to previous solution, incubated for 60 min, the absorbance was measured at 405 nm and the following equation was used for evaluating the percentage enzyme inhibition: Inhibition (%) = [(A_blank_ − (A_sample_ − A_correction_)/(A_blamk_)] × 100; where A_blank_ is the absorbance of the control. A_sample_ and A_correction_ were the absorbance of the sample with and without the DTNB, respectively.

#### 3.9.3. Lipoxygenase Inhibition Assay

The 5-lipoxygenase assay was carried out according to the slightly modified method of El Guendouz et al. [66]. Fifteen µL of enzyme solution (54 mg/mL borate buffer, 0.005% Tween 20, pH = 9), 937 µL borate buffer, 20 µL of various concentrations of VOs, and 50 µL of linoleic acid (1 mM) were mixed, the absorbance was measured at 243 nm and the percentage enzyme inhibition was calculated using the formula: Inhibition (%) = [(A_blank_ − (A_sample_ − A_correction_)/(A_blamk_)] × 100; where A_blank_ is the absorbance of the control. A_sample_ and A_correction_ were the absorbance of the sample with and without the enzyme, respectively.

### 3.10. Statistical Analysis

All the data were carried out in three replicates and presented as means  ±  standard deviation (SD). All the data collects were normally distributed, so the independent samples Student’s *t*-test and Tukey’s post hoc test were adjusted to estimate for significant differences between groups’ means. The statistical were conducted using Minitab^®^ 17.1.0 program (LEADTOOLS © 1991–2004, LEAD Technologies, Inc., Charlotte, NC, USA).

## 4. Conclusions

*Rosmarinus officinalis*, *Lavandula angustifolia*, and *Origanum compactum* volatile oils (VOs) obtained by simultaneous hydro- and steam-distillation assisted by carbon dioxide (SHSDACD) attained higher yields with an extraction time < 30 min compared to simultaneous hydro- and steam-distillation (SHSD), consuming less energy (<10 times lower than SHSD) due to the enhancement noticed in the K distillation rate constant.

Some differences between SHSD and SHSDACD oregano VOS chemical composition were observed, nevertheless it was unclear the reason for such dissimilarity. The biological properties of SHSD and SHSDACD VOs were distinct depending on the plant species and on the extraction process. Generally, the SHSDACD VOs presented better activities than those obtained by SHSD.

In this comparative study, SHSDACD can be classified as a green method, since it was possible to reduce the extraction time of the volatile compounds and, therefore, the energy consumption, without loss of quality.

## Figures and Tables

**Figure 1 pharmaceuticals-15-00567-f001:**
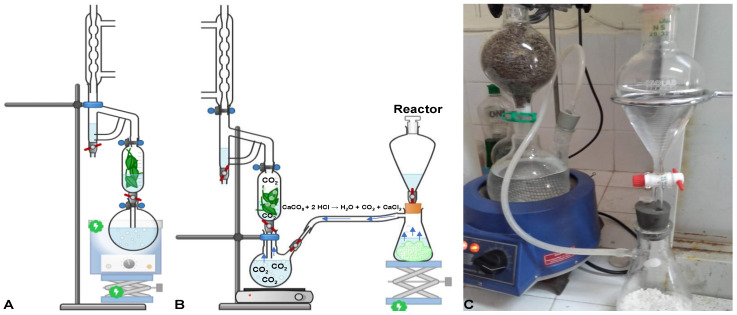
Schematic representation of the simultaneous hydro- and steam-distillation (SHSD) (**A**) and SHSD assisted by carbon dioxide (SHSDACD); (**B**) Detail of the in-lab built system of SHSDACD (**C**).

**Figure 2 pharmaceuticals-15-00567-f002:**
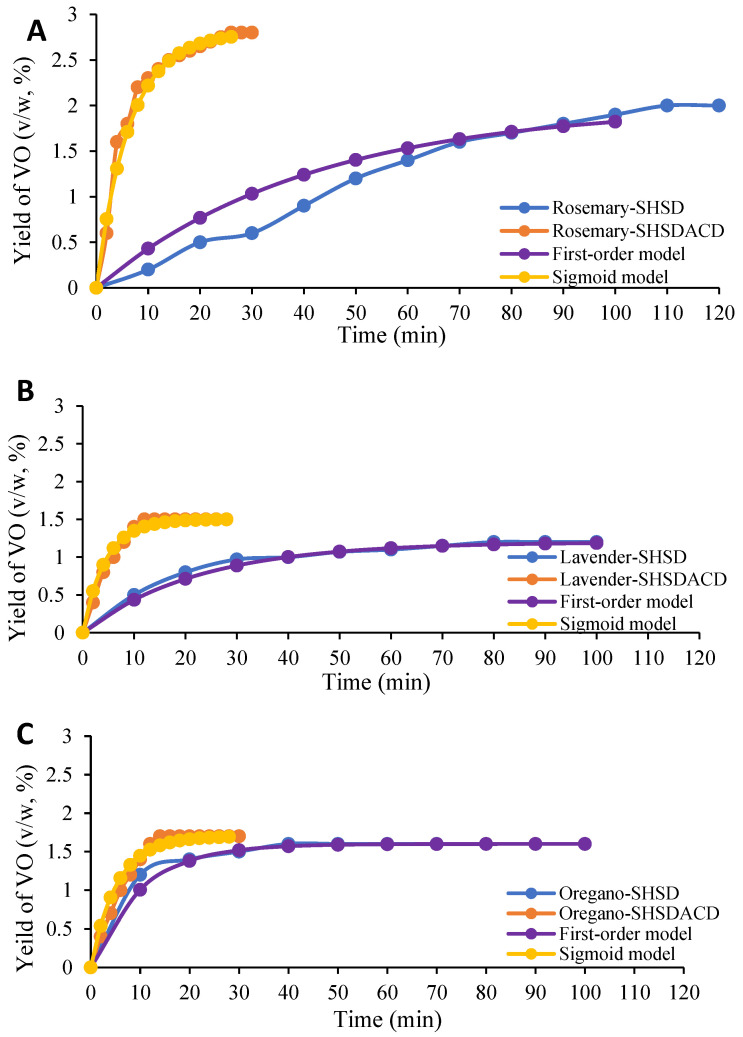
VOs yields’ kinetic curves for *R. officinalis* (**A**); *L. angustifolia* (**B**); and *O. compactum* (**C**) obtained by simultaneous hydro- and steam-distillation (SHSD) and SHSD assisted by carbon dioxide (SHSDACD). Points lines represent the actual data, and the fitting behavior predicted by first order and sigmoid kinetic models.

**Figure 3 pharmaceuticals-15-00567-f003:**
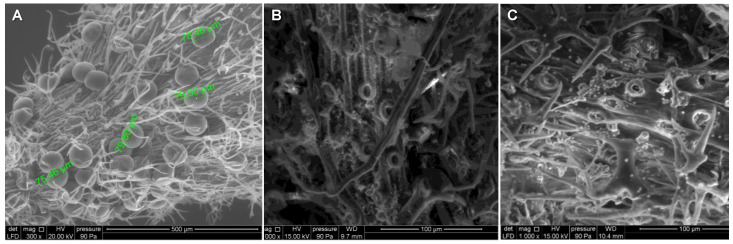
Representative scanning electron micrographs (SEM) of *Lavandula angustifolia* flowers before distillation (**A**) and after simultaneous hydro- and steam-distillation (SHSD) (**B**) and SHSD assisted by carbon dioxide (SHSDACD) (**C**).

**Figure 4 pharmaceuticals-15-00567-f004:**
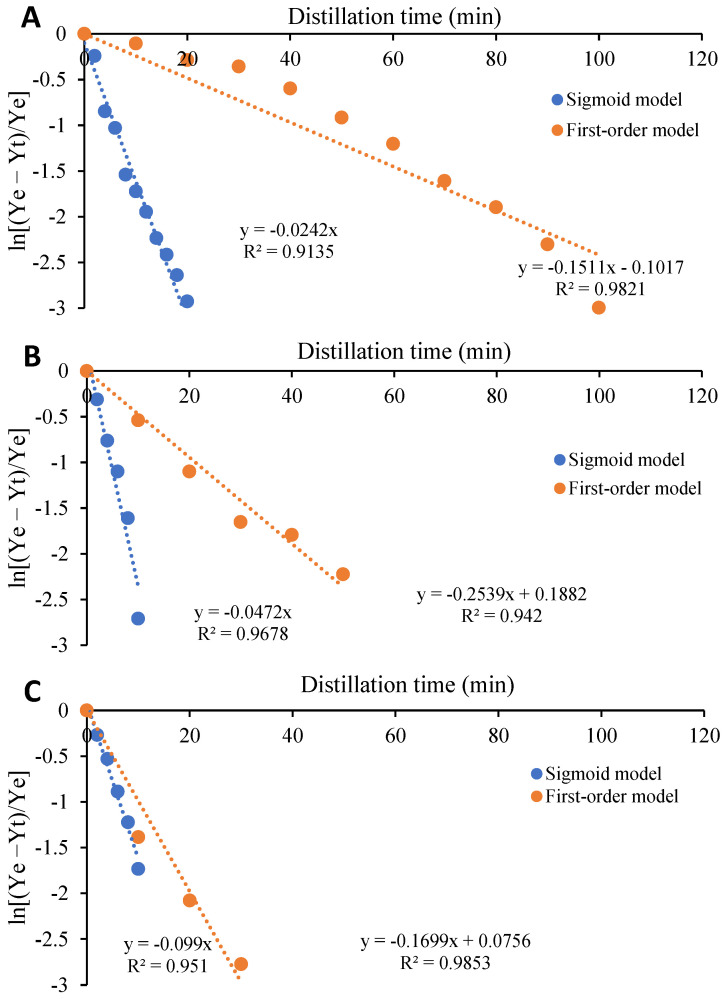
Dependence of ln [(Y_e_ − Y_t_)/Y_e_] with distillation time. (**A**) *R. officinalis;* (**B**) *L. angustifolia;* and (**C**) *O. compactum*.

**Figure 5 pharmaceuticals-15-00567-f005:**
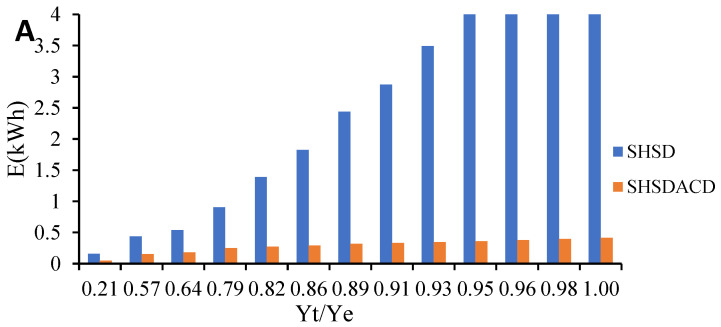
Comparison of energy consumption during VOs isolation by simultaneous hydro- and steam-distillation (SHSD) and SHSD assisted by carbon dioxide (SHSDACD). (**A**) *R. officinalis;* (**B**) *L. angustifolia;* and (**C**) *O. compactum*.

**Table 1 pharmaceuticals-15-00567-t001:** Parameters of kinetic model of Equations (2)–(4), constant and coefficient of extraction, extraction error of the mean squared error (MSE) and *R. officinalis, L. angustifolia*, and *O. compactum* VOs yields obtained with simultaneous hydro- and steam-distillation (SHSD) and SHSD assisted by carbon dioxide (SHSDACD).

Extraction Methods	Samples	Extraction Time	P	Plant Mass	Y_∞_	First-Order Model			Adsorption Model			Sigmoid Model			
						K	MSE	R^2^	b	MSE	R^2^	K	b	MSE	R^2^
		(min)	(W)	(g)	(%, *v*/*w*)	(min^−1^)	(%)	(%)		(%)	(%)	(min^−1^)		(%)	(%)
SHSD	*R. officinalis*	120	220	100	2.00 ± 0.01 *^b^*	0.02	4	99.4	77.6	84	63.2	0.01	0.02	98	55.3
	*L. angustifolia*	100	220	100	1.20 ± 0.00 *^ef^*	0.04	1	98.0	12.1	7	50.0	0.04	0.13	10	75.0
	*O. compactum*	60	220	100	1.60 ± 0.05 *^cd^*	0.10	0	98.9	3.14	6	96.6	0.10	0.02	1	98.3
SHSDACD	*R. officinalis*	30	220	100	2.80 ± 0.10 ^*a*^	0.16	6	99.3	2.73	86	56.3	0.15	−0.22	0	93.9
	*L. angustifolia*	28	220	100	1.50 ± 0.00 *^e^*	0.23	4	92.6	4.60	7	27.7	0.25	−0.21	0	99.8
	*O. compactum*	20	220	100	1.70 ± 0.01 *^c^*	0.19	1	99.2	2.73	5	94.7	0.27	−0.26	0	99.2

Results are average value ± SD of triplicate measurement. Values in the same column followed by the same superscript letter are not significantly different (*p* > 0.05) by Tukey’s multiple range test.

**Table 2 pharmaceuticals-15-00567-t002:** Percentage composition of *R. officinalis* (Ro), *L. angustifolia* (La), and *O. compactum* (Oc) VOs, obtained by simultaneous hydro- and steam-distillation (SHSD) and SHSD assisted by carbon dioxide (SHSDACD).

Components	RI ^a^	RI ^b,c^	SHSD			SHSDACD		
			Ro	La	Oc	Ro	La	Oc
α-Thujene	924	929	0.2		0.4	t	t	t
α-Pinene	930	931	10.8	0.4	0.6	9.9	0.3	0.3
Camphene	938	938	3.4	0.5	0.1	2.6	0.5	0.1
Thuja-2,4(10)-diene *	940	939	t	t		t	t	
Sabinene	958	959	0.5	t		0.7	t	
1-Octen-3-ol	961	972	t	t	1.0	t	t	2.0
3-Octanone	961	962			1.0			2.0
β-Pinene	963	962	1.0	0.7		1.4	0.6	
Dehydroxy-*trans*-linalool oxide *	973			0.1			t	
Dehydro-1,8-cineole	973	972			t			0.1
β-Myrcene	975	980	1.0	0.5	1.1	1.1	0.1	0.7
Dehydroxy-*cis*-linalool oxide *	995			0.1			0.1	
Hexyl acetate	995	995		0.1			t	
α-Phellandrene	995	986	t		0.1	0.2		t
α-Terpinene	1002	1001	0.1		1.2	0.4		0.2
*p*-Cymene	1003	1004	2.5	0.3	27.3	1.9	0.3	25.0
1,8-Cineole	1005	1010	50.2	16.1	0.7	48.2	13.1	1.5
β-Phellandrene	1005	1011			t			t
Limonene	1009	1014	3.4	0.7	0.2	2.2	0.7	t
*cis*-β-Ocimene	1017	1015	t	0.1	t	t	t	t
*trans*-β-Ocimene	1027	1026		0.3	t		t	t
γ-Terpinene	1035	1043	t	0.1	9.8	0.4	t	0.8
*trans*-Sabinene hydrate	1037	1096			0.2			0.2
*cis*-Linalool oxide (furanoid)	1045			2.3			2.3	
*trans*-Linalool oxide (furanoid)	1059			1.8			1.8	
2,5-Dimethyl styrene	1059	1076	0.1		0.1	t		0.2
Terpinolene	1064	1077	t	0.1	t	0.2	t	t
*cis*-Sabinene hydrate	1066	1080			0.2			0.3
Linalool	1074	1082	1.1	20.5	1.5	1.0	14.2	2.6
*endo*-Fenchol	1085	1088	0.1			t		
1-Octen-3-yl acetate	1086	1092		0.8			0.8	
α-Campholenal	1092	1103	0.1			t		
Camphor	1102	1107	15.3	16.5	0.2	11.7	15.1	0.4
Hexyl isobutanoate	1127	1130		0.1			0.1	
Nerol oxide	1127	1140		0.2			0.3	
Borneol	1134	1147	3.7	10.4	0.4	5.2	10.3	0.8
Cryptone *	1143	1148		0.4			0.3	
Terpinen-4-ol	1148	1153	0.9	0.9	0.9	0.9	0.8	0.9
*p*-Cymen-8-ol	1148	1156			0.3			0.9
α-Terpineol	1159	1157	3.8	2.7	3.3	4.3	0.8	19.3
Hexyl butyrate	1173	1174		0.3			0.2	
Bornyl formate	1200	1208		0.2			0.2	
Cumin aldehyde	1200	1221		0.2			0.2	
Thymoquinone	1210	1226			0.1			0.4
Methyl thymol	1210	1227			0.7			4.8
Hexyl 2-methyl butyrate	1220	1222		t			t	
Geraniol	1236	1234		1.2			t	
Linalyl acetate	1245	1255	0.2	13.1	t	t	25.5	t
Bornyl acetate	1265	1259	0.2	0.2		0.3	0.2	
Cumin alcohol	1265	1265			0.6			0.7
Thymol	1275	1278	0.1		37.8	0.1		10.5
Lavandulyl acetate	1278	1273		0.8			0.8	
Carvacrol	1286	1278	t		5.5	t		17.9
Myrtenyl acetate	1290	1285		0.1			0.5	
Hexyl tiglate	1316	1310		0.1			0.1	
Eugenol	1327	1327	0.2			t		
α-Terpenyl acetate	1334	1334		0.5			0.1	
Geranyl acetate	1370	1360	t	1.0		t	0.1	
β-Caryophyllene	1414	1415/*1421*	0.1	0.3	1.1	5.6	0.7	1.1
α-Santalene	1422	*1422*		0.1			0.3	
Geranyl acetone	1434	1426	t			t		
α-Humulene	1447	1439/*1455*	t			0.9		0.1
*trans*-α-Farnesene	1455	*1446*		0.1			0.4	
β-Bisabolene	1500	1487	t		0.1	t		0.1
γ-Cadinene	1500	*1507*		0.4			1.1	
β-Sesquiphellandrene	1508	1508			t			0.1
*tert*-Butylhydroquinone	1510				t			0.8
α-Calacorene	1525	1525/*1527*	t			t		
β-Caryophyllene oxide	1561	1565	0.2	0.8	1.6	0.3	1.5	1.5
Humulene epoxide	1580	1581			t			0.1
T-Cadinol	1616	*1540*	0.2	1.4		t	1.9	
α-Bisabolol oxide B *	1630	1616		0.1			0.1	
α-Bisabolol	1656	1656	t	0.5		t	0.6	
**% Identification**			99.4	98.1	98.1	99.5	97.0	96.4
**Grouped components**								
Monoterpene hydrocarbons			23.0	3.7	40.9	21.0	2.5	27.3
Oxygen-containing monoterpenes			75.7	89.3	52.4	71.7	86.7	61.3
Sesquiterpene hydrocarbons			0.1	0.9	1.2	6.5	2.5	1.4
Oxygen-containing sesquiterpenes			0.4	2.8	1.6	0.3	4.1	1.6
Phenylpropanoid			0.2			t		
Others			t	1.4	2.0	t	1.2	4.8

RI ^a^: In-lab calculated retention index relative to C_9_-C_17_
*n*-alkanes on the DB-1column. * Identification based on mass spectra only. STDEV < 1%; RI ^b,c^, Regular font values from [32], italic values from [33]; RI ^b^, literature retention indices on DB-1 or similar phase column (100% Dimethylpolysiloxane) not from the authors’ lab; RI ^c^, literature retention indices on a Cp-Sil 5 (100% Dimethylpolysiloxane). No RI value: no RI was found on the database, either because it was not present, it was from the authors’ lab, or it was from different column type and/or oven program.

**Table 3 pharmaceuticals-15-00567-t003:** Antioxidant activity of *R. officinalis, L. angustifolia*, and *O. compactum* VOs obtained by simultaneous hydro- and steam-distillation (SHD) and SHD assisted by carbon dioxide (SHDACD).

Extraction Methods	VOs Samples	Half Maximal Inhibitory Concentration IC_50_ (mg/mL)		
		DPPH	NO	Superoxide
SHD	*R. officinalis*	3.06 ± 0.23 *^b^*	ND	ND
	*L. angustifolia*	4.92 ± 0.20 *^a^*	ND	1.53 ± 0.07 *^a^*
	*O. compactum*	1.51 ± 0.03 *^c^*	0.20 ± 0.00 *^c^*	1.40 ± 0.12 *^ab^*
SHDACD	*R. officinalis*	1.09 ± 0.05 *^d^*	4.23 ± 0.24 *^b^*	0.77 ± 0.02 *^c^*
	*L. angustifolia*	3.50 ± 0.13 *^b^*	5.02 ± 0.25 *^a^*	0.21 ± 0.01 *^e^*
	*O. compactum*	0.01± 0.00 *^e^*	0.05 ± 0.00 *^d^*	0.55 ± 0.02 *^d^*

ND: not determined (*R. officinalis*: IP_NO_ = 21.14 ± 1.18% at 22.25 mg/mL; IP _superoxide_ = 10.67 ± 084% at 7.42 mg/mL) (*L. angustifolia*: IP_NO_ = 29.36 ± 0.26% at 21.13 mg/mL). Values in the same column followed by the same superscript letter are not significantly different (*p* > 0.05) by Tukey’s multiple range test.

**Table 4 pharmaceuticals-15-00567-t004:** Enzymatic inhibitory activity of *R. officinalis, L. angustifolia,* and *O. compactum* VOs obtained by simultaneous hydro- and steam-distillation (SHSD) and SHSD assisted by carbon dioxide (SHSDACD).

Extraction Methods	VOs Samples	Half Maximal Inhibitory Concentration IC_50_ (µg/mL)		
		α-Glucosidase	Acetylcholinesterase	Lipoxygenase
SHD	*R. officinalis*	219.44 ± 2.53 *^c^*	513.58 ± 33.13 *^a^*	819.97 ± 15.12 *^a^*
	*L. angustifolia*	721.07 ± 83.06 *^a^*	499.19 ± 14.45 *^ab^*	143.78 ± 2.59 *^b^*
	*O. compactum*	107.18 ± 2.11 *^d^*	10.66 ± 0.45 *^f^*	123.60 ± 5.37 *^c^*
SHDACD	*R. officinalis*	8.57 ± 0.39 *^e^*	349.16 ± 16.76 *^c^*	21.45 ± 0.22 *^ef^*
	*L. angustifolia*	459.62 ± 30.21 *^b^*	83.35 ± 4.01 *^d^*	31.33 ± 0.57 *^d^*
	*O. compactum*	3.24 ± 0.30 *^f^*	71.47 ± 2.77 *^e^*	25.08 ± 0.49 *^ef^*

Results are average value ± SD of triplicate measurement. Values in the same column followed by the same superscript letter are not significantly different (*p* > 0.05) by Tukey’s multiple range test.

## Data Availability

Data is contained within the article.

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
