# Peer review of "Unassisted and Carbon Dioxide-Assisted Hydro- and Steam-Distillation: Modelling Kinetics, Energy Consumption and Chemical and Biological Activities of Volatile Oils"

_pharmaceuticals, 2022, doi:10.3390/ph15050567_

Round 1

Reviewer 1 Report

Dear Authors,

The manuscript is well written, is easy to follow, and the experimental data are presented properly. Scientific novelty of the obtained results is beyond doubt. I find this manuscript interesting, detailed and scientifically justified.

The research is extensive and covers different methods of extractions (hydro- and steam-distillation (SHSD); and SHSD assisted by carbon dioxide, using adopted modelling approach), analytical determination of isolated compounds, examining of various biological activities, namely antioxidant, anti-glucosidase, anti-acetylcholinesterase, and anti-inflammatory properties followed by discussion and analysis of data. Additionally, the present study estimated kinetic extraction and energy consumption of these two extraction techniques.

In order to improve the manuscript, the following suggestions should be taken into account by the authors:

Please emphasize originality of the method of hydro- and steam-distillation assisted by CO2 (SHSDACD). Does this extraction method could be applied in industry?

Specify lavender herbal material. According to the E. Bejar, appreciable amounts of camphor are characteristic for spike lavender (Lavandula latifolia), but not for the oils of Lavandula angustifolia.

 E. Bejar. Adulteration of English Lavender (Lavandula angustifolia) essential oil. Botanical Adulterants Prevention Bulletin. Austin, TX: ABC-AHP-NCNPR Botanical Adulterants Prevention Program; 2020.

 Lines 196-225: Exact quantities, as presented in the Table 2 (means of percentage with one number after dot) of constituents have to be used in the text.

Lines 444-445: Please indicate range of concentrations used for DPPH test and IC50 values.

After this minor revision, I recommend the present manuscript for publication. 

Author Response

Dear Editors

Prof. Dr. Antonio Eduardo Miller Crotti

Prof. Dr. Eliane de Oliveira Silva

Date: 05/04/2022

Subject: Reply Letter

Dear Professors

We thank you for your e-mail of 29 March 2022 with the Decision Letter on Pharmaceuticals Manuscript ID: pharmaceuticals-1660547 entitled "Unassisted and carbon dioxide-assisted hydro- and steam distillation: modelling kinetics, energy consumption and chemical and biological activities of volatile oils".

We have read it carefully and we do understand most of the referee’s comments. Please find below our reply and comments addressing each point raised by the reviewers. All corrections, additions and changes performed in the MS text have been marked in blue.

Reviewer 1

The manuscript is well written, is easy to follow, and the experimental data are presented properly. Scientific novelty of the obtained results is beyond doubt. I find this manuscript interesting, detailed and scientifically justified.

The research is extensive and covers different methods of extractions (hydro- and steam-distillation (SHSD); and SHSD assisted by carbon dioxide, using adopted modelling approach), analytical determination of isolated compounds, examining of various biological activities, namely antioxidant, anti-glucosidase, anti-acetylcholinesterase, and anti-inflammatory properties followed by discussion and analysis of data. Additionally, the present study estimated kinetic extraction and energy consumption of these two extraction techniques.

In order to improve the manuscript, the following suggestions should be taken into account by the authors:

Please emphasize originality of the method of hydro- and steam-distillation assisted by CO2 (SHSDACD). Does this extraction method could be applied in industry?

SHSDACD is basically an alternative to conventional hydrodistillation wherein the plant materials are subjected to a flux of water vapor mixed with carbon dioxide gas generated and diffused trough the plants. This innovative process reduces the extraction time and the energy consumption, improving simultaneous the quantity and the quality of volatile oils extracts. The main goal of the study was to find a cheap process for obtaining volatiles without loss of quality, which needs to be assayed in larger scale. At the moment, it is difficult for us to answer with certitude that this procedure can easily be applied in industry.

Specify lavender herbal material. According to the E. Bejar, appreciable amounts of camphor are characteristic for spike lavender (Lavandula latifolia), but not for the oils of Lavandula angustifolia.

  1. Bejar. Adulteration of English Lavender (Lavandula angustifolia) essential oil. Botanical Adulterants Prevention Bulletin. Austin, TX: ABC-AHP-NCNPR Botanical Adulterants Prevention Program; 2020.

Lavandula latifolia is characterized by higher percentages of camphor and 1,8-cineole, as observed in the present work, nevertheless with lower amounts of linalool as is common in L. latifolia (34-50%) according to the same reference cited by the reviewer. However, the chemical profile is much closer to L. latifolia volatiles than to the L. angustifolia volatiles, whereby, we will consider Lavandula spp. instead of L. angustifolia or L. latifolia. All terms L. angustifolia were replaced by Lavandula spp.

 Lines 196-225: Exact quantities, as presented in the Table 2 (means of percentage with one number after dot) of constituents have to be used in the text.

As requested, the values depicted in the Table are now cited along the text.

Lines 444-445: Please indicate range of concentrations used for DPPH test and IC50 values.

The concentrations were added in the manuscript (blue colour)

Reviewer 2

In the present study, two methods are used for the extraction of essential oils: hydrodistillation (SHSD) and assisted by carbon dioxide distillation (SHSDACD), the kinetic extraction and the energy consumption are compared. In addition, the difference in yield, chemical composition, and biological activities (antioxidant, anti-glucosidase, anti-acetylcholinesterase and anti-inflammatory) is analyzed.

 Considering that the "carbon dioxide-assisted distillation" method used in the present study is new in the extraction of essential oils and that significantly better results are achieved compared to traditional methods, the "SHSDACD" method should be better detailed in the Introduction and in Materials and Methods, a photograph of the equipment should also be included.

The photograph of the process SHSDACD was added and now is Figure 1C

Title

- Consider changing the word "distillation" to "hydrodistillation"

The term was changed.

Abstract

- Define if it is "hydro" or "steam" distillation, I consider that since it cannot be ensured that the steam is always in a state of saturation, it should be considered only as hydrodistillation

Generally, it is considered that steam distillation is as a process where the plants are exposed to saturated steam, obtained by water vapour situated below the plant material. This steam is able to heat the plant within the biomass flask, with condensation and convection, and then conduction, allowing the release of the volatile’s compounds (table of saturated steam temperature and pressure, https://www.tlv.com/global/ME/steam-theory/how-to-read-a-steam-table.html#toc_1).

The term simultaneous hydro and steam distillation were used due to the fact that the process used in our work is a combined of the two conventional methods, hydrodistillation and steam distillation, where the aromatic plants in the biomass flask are heated by saturated steam and carbon dioxide at boiling point. After condensation of the first water cycle. The refluxed excess of water with Clevenger extractor saturated the aromatic plants in the biomass flask before dropping to the lower distiller flask with water, in the state the plants are quasi-submerged in water, and that why we used the term ’hydro-’. After this first cycle, the whole system in uniform condition of liquid-solid and gas-solid extraction.

-O. compactum, escribir nombre completo

  1. was replaced by Origanum as requested.

Introduction

 The hypothesis, "that this system would allow more efficient and homogeneous internal diffusion through the material, accelerating the extraction process as well as helping to achieve a low extraction time and energy" should be explained or supported by bibliography.

This was added and is in blue colour 

Results and Discussion

- In figure 2B, the SHSDACD curve does not present a good fit of the model, however in table 1 an R of 99.8 is indicated.

Thank you for pointing out that and we apologize for that error, it was the wrong graphic. The correct one replaced the wrong graphic.

-Table 1. It is maintained that the experiments were carried out in triplicate, but standard deviations of 0.00 are indicated, due to the oiliness of the essential oil, the error in the measuring instruments, and other particularities of the same process that is not possible.

The experiment was conducted in the same conditions, repeatability conditions, at the same day, in a way to minimize the error which is not related to the instrument but to the experimenter himself. We used the same apparatus as for the condition where the same:100 g of plant material of the same lot with 17 % humidity, 1000 mL of water, 100°C temperature.

- Line 134-135, R values greater than 98% should be considered a good fit.

Yes, but if we used R values greater than 98% we will exclude the good fit to rosemary data with  R²= 93.9%

- Table 2. The RI of the bibliography and the standard deviation of the relative

abundance (%) of each compound must be included.

  The RI of the bibliography was added in the Table and, in order not to overcrowd the table the information on the standard deviation was included in the table footnote as being always below 1%.

-Table 4. It is suggested to express in ug/mL.

This was done.

-Table 4., O. compactum has a value of 0.00?

As values are presented in ug/mL, now the value is not 0.00 mg/mL, but 2 ug/mL

Materials and Methods

-Line 346, paragraph must start with full word.

This was done.

-In reference 54 the SHSD Method is not described.

The sentence was rephrased and the reference removed

-The CO2 flow should be indicated and related to the distillate flow (condensate, water + oil), this parameter is important for the reproducibility of the process.

It was added.

-Equation 7, Should it be Ye?

It was replaced

-3.7. Chemical analysis of volatile oils. At least the type of equipment, the columns and the form of identification of the compounds must be indicated.

It was added.

-Rewrite or delete the statement "SHSDACD can be considered as an adequate extraction

It was deleted.

process for obtaining VOs with good anti-inflammatory ability by inhibiting the lipoxygenase activity, ....", in the study oils this is true, but in other oils this It is uncertain.

It was deleted.

Conclusions

-Paragraph 2 (line 521-523) is not a conclusion, it seems more like an abstract.

It was rewritten.

We hope that we have adequately addressed all the reviewers’ remarks and questions, and that the manuscript is now suitable for publication.

Yours sincerely

Maria da Graça Miguel

Reviewer 2 Report

In the present study, two methods are used for the extraction of essential oils: hydrodistillation (SHSD) and assisted by carbon dioxide distillation (SHSDACD), the kinetic extraction and the energy consumption are compared. In addition, the difference in yield, chemical composition, and biological activities (antioxidant, anti-glucosidase, anti-acetylcholinesterase and anti-inflammatory) is analyzed.

Considering that the "carbon dioxide-assisted distillation" method used in the present study is new in the extraction of essential oils and that significantly better results are achieved compared to traditional methods, the "SHSDACD" method should be better detailed in the Introduction and in Materials and Methods, a photograph of the equipment should also be included.

Title

- Consider changing the word "distillation" to "hydrodestilation"

Abstract

- Define if it is "hydro" or "steam" distillation, I consider that since it cannot be ensured that the steam is always in a state of saturation, it should be considered only as hydrodistillation

-O. compactum, escribir nombre completo

Introduction

 The hypothesis, "that this system would allow more efficient and homogeneous internal diffusion through the material accelerating the extraction process as well as helping to achieve a low extraction time and energy" should be explained or supported by bibliography.

Results and Discussion

- In figure 2B, the SHSDACD curve does not present a good fit of the model, however in table 1 an R of 99.8 is indicated.

-Table 1. It is maintained that the experiments were carried out in triplicate, but standard deviations of 0.00 are indicated, due to the oiliness of the essential oil, the error in the measuring instruments, and other particularities of the same process that is not possible.

- Line 134-135, R values greater than 98% should be considered a good fit.

- Table 2. The RI of the bibliography and the standard deviation of the relative abundance (%) of each compound must be included.

-Table 4. It is suggested to express in ug/mL.

-Table 4., O. compactum has a value of 0.00?

Materials and Methods

-Line 346, paragraph must start with full word.

-In reference 54 the SHSD Method is not described.

-The CO2 flow should be indicated and related to the distillate flow (condensate, water + oil), this parameter is important for the reproducibility of the process.

-Equation 7, Should it be Ye?

-3.7. Chemical analysis of volatile oils. At least the type of equipment, the columns and the form of identification of the compounds must be indicated.

-Rewrite or delete the statement "SHSDACD can be considered as an adequate extraction process for obtaining VOs with good anti-inflammatory ability by inhibiting the lipoxygenase activity, ....", in the study oils this is true, but in other oils this It is uncertain.

Conclusions

-Paragraph 2 (line 521-523) is not a conclusion, it seems more like an abstract.

Author Response

(The authors gave the same response as above.)

Round 2

Reviewer 2 Report

I thank the authors for their efforts to improve their manuscript and confirm that the paper can be published in its current form.

Author Response

Dear Editors

Prof. Dr. Antonio Eduardo Miller Crotti

Prof. Dr. Eliane de Oliveira Silva

Date: 21/04/2022

Subject: Reply Letter

Dear Professors

We thank you for your e-mail of 19 April 2022 with the Decision Letter on Pharmaceuticals Manuscript ID: pharmaceuticals-1660547 entitled "Unassisted and carbon dioxide-assisted hydro- and steam distillation: modelling kinetics, energy consumption and chemical and biological activities of volatile oils".

We have read it carefully and we do understand most of the referee’s comments. Please find below our reply and comments addressing each point raised by the reviewers. All corrections, additions and changes performed in the MS text have been marked in blue.

First, I have some concerns about the identification of Lavandula species from which the authors extracted the essential oil. In the first version of the manuscript, the authors classified this species as Lavandula angustifolia. Nevertheless, to attend to one of the reviewers’ comments, the authors changed the classification to Lavandula spp. This led us to believe that the botanical classification was not made by a botanist. Because the authors had the vouchers, I do not think it will be a big deal to check the correct identification with a botanist.

Your concern is very correct and indeed we were waiting for receiving your reply to request a correction. Meanwhile, during the time the manuscript was on the second revision we confirmed with the botanist that classified and keep the voucher, that the material is indeed Lavandula angustifolia. So, despite the chemical variability we would like to change to the correct botanical name, according to the taxonomic classification (this was marked in blue font). Our apologies for the confusion caused.

 I also noticed that some of the retention index values from the literature were not included in Table 2. Is there any special reason for that? Is this compound new in the literature? It will be suitable to include at least a footnote explaining why these RI values did not appear in Table 2.

An additional sentence was added (marked in blue font). The compounds are not new, but either there is no RI on the database or, if there is, it is from a different column, it is not linear retention, or it is from the authors lab and it is strange to insert data from the authors lab.

 I also find come experimental RI that differ significantly from the RI from literature, like trans-sabinene hydrate (RIexp = 1037; RIliter = 1096), 2,5-dimethyl styrene (RIexp = 1059; RIlit = 1076), cis-sabinene hydrate (RIexp = 1066; RIlit = 1080), cumin aldehyde (RIexp – 1200; RIlit = 1221). In the case of beta-caryophyllene,  alpha-humulene, and alpha-calacorene two RI values from the literature were provided and the reason why the authors did that is not clear.

True. The problem is the same as above. The data from the database must be from a similar column, and be linear retention index, and it is very difficult to find always from the same oven program for such an extensive table. Thus, although we try to use as much as possible the same authors as reference, sometimes this is not possible, and some difference in values may occur. Nevertheless, they are always within the same alkane range. So, this just reflects different operative conditions.

 We hope that we have adequately addressed all the reviewers’ remarks and questions, and that the manuscript is now suitable for publication.

Yours sincerely

Maria da Graça Miguel
